# Learning Neural Event Functions for Ordinary Differential Equations

**Ricky T. Q. Chen**[*]
University of Toronto; Vector Institute
rtqichen@cs.toronto.edu

**Brandon Amos, Maximilian Nickel**
Facebook AI Research
{bda,maxn}@fb.com

## Abstract

The existing Neural ODE formulation relies on an explicit knowledge of the termination time. We extend Neural ODEs to implicitly defined termination criteria modeled by *neural event functions*, which can be chained together and differentiated through. Neural Event ODEs are capable of modeling discrete and instantaneous changes in a continuous-time system, without prior knowledge of when these changes should occur or how many such changes should exist. We test our approach in modeling hybrid discrete- and continuous- systems such as switching dynamical systems and collision in multi-body systems, and we propose simulation-based training of point processes with applications in discrete control.

## 1 Introduction

Event handling in the context of solving ordinary differential equations (Shampine & Thompson, 2000) allows the user to specify a termination criteria using an event function. Part of the reason is to introduce discontinuous changes to a system that cannot be modeled by an ODE alone. Examples being collision in physical systems, chemical reactions, or switching dynamics (Ackerson & Fu, 1970). Another part of the motivation is to create discrete outputs from a continuous-time process; such is the case in point processes and event-driven sampling (*e.g.* Steinbrecher & Shaw (2008); Peters et al. (2012); Bouchard-Côté et al. (2018)). In general, an event function is a tool for monitoring a continuous-time system and performing instantaneous interventions when events occur.

The use of ordinary differential equation (ODE) solvers within deep learning frameworks has allowed end-to-end training of Neural ODEs (Chen et al., 2018) in a variety of settings. Examples include graphics (Yang et al., 2019; Rempe et al., 2020; Gupta & Chandraker, 2020), generative modeling (Grathwohl et al., 2018; Zhang et al., 2018; Chen & Duvenaud, 2019; Onken et al., 2020), time series modeling (Rubanova et al., 2019; De Brouwer et al., 2019; Jia & Benson, 2019; Kidger et al., 2020), and physics-based models (Zhong et al., 2019; Greydanus et al., 2019).

However, these existing models are defined with a fixed termination time. To further expand the applications of Neural ODEs, we investigate the parameterization and learning of a termination criteria, such that the termination time is only implicitly defined and will depend on changes in the continuous-time state. For this, we make use of event handling in ODE solvers and derive the gradients necessarily for training event functions that are parameterized with neural networks. By introducing differentiable termination criteria in Neural ODEs, our approach allows the model to efficiently and automatically handle state discontinuities.

### 1.1 Event Handling

Suppose we have a continuous-time state $z(t)$ that follows an ODE $\frac{dz}{dt} = f(t, z(t), \theta)$—where $\theta$ are parameters of $f$—with an initial state $z(t_0) = z_0$. The solution at a time value $\tau$ can be written as

$$\texttt{ODESolve}(z_0, f, t_0, \tau, \theta) \triangleq z(\tau) = z_0 + \int_{t_0}^{\tau} f(t, z(t), \theta) \, dt. \tag{1}$$

In the context of a Neural ODE, $f$ can be defined using a Lipschitz-continuous neural network. However, since the state $z(t)$ is defined through infinitesimal changes, $z(t)$ is always continuous in

---

[*]Work done while at Facebook AI Research.

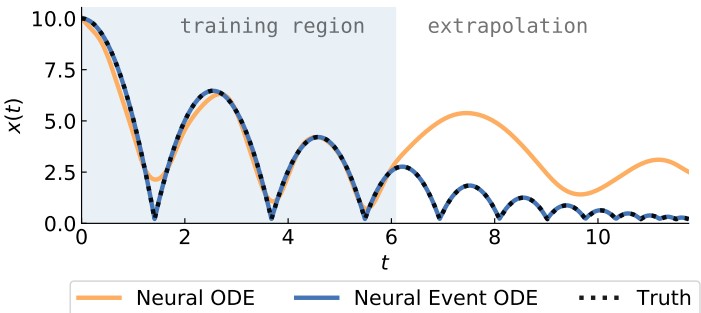

Figure 1: Dynamics of a bouncing ball can be recovered by a Neural Event ODE. Meanwhile, a non-linear Neural ODE has trouble modeling sudden changes and performs poorly at extrapolation.

$t$. While smooth trajectories can be a desirable property in some settings, trajectories modeled by an ODE can have limited representation capabilities (Dupont et al., 2019; Zhang et al., 2020) and in some applications, it is desirable to model *discontinuities* in the state.

**Bouncing ball example**     As a motivating example of a system with discontinuous transitions, consider modeling a bouncing ball with classical mechanics. In an environment with constant gravity, a Markov state for representing the ball is a combination of position $x(t) \in \mathbb{R}$ and velocity $v(t) \in \mathbb{R}$,

$$z(t) = [x(t), v(t)], \qquad \frac{dz(t)}{dt} = [v(t), a], \qquad (2)$$

where $a$ is a scalar for acceleration, in our context a gravitational constant.

To simulate this system, we need to be mindful that the ball will eventually pass through the ground—when $x(t) \leq r$ for some $r$ that is the radius of the ball—but when it hits the ground, it bounces back up. At the moment of impact, the sign of the velocity is changed instantaneously. However, no such ODE can model this behavior because $v(t)$ needs to change discontinuously at the moment of impact. This simple bouncing ball is an example of a scenario that would be ill-suited for a Neural ODE alone to model.

In order to model this discontinuity in the state, we can make use of *event functions*. Event functions allow the ODE to be terminated when a criteria is satisfied, at which point we can instantaneously modify the state and then resume solving the ODE with this new state. Concretely, let $g(t, z(t), \phi)$ be an event function with $\phi$ denoting a set of parameters. An ODE solver with event handling capabilities will terminate at the first occurrence when the event function crosses zero, *i.e.* time $t^*$ such that $g(t^*, z(t^*), \phi) = 0$, conditioned on some initial value. We express this relationship as

$$t^*, z(t^*) = \texttt{ODESolveEvent}(z_0, f, g, t_0, \theta, \phi). \qquad (3)$$

Note that in contrast to eq. (1), there is no predetermined termination time. The time of termination $t^*$ has to be solved alongside the initial value problem as it depends on the trajectory $z(t)$. Nevertheless, `ODESolveEvent` strictly generalizes `ODESolve` since the event function can simply encode an explicit termination time and is reduced back into an `ODESolve`. The benefits of using `ODESolveEvent` lie in being able to define event functions that depend on the evolving state.

Going back to the bouncing ball example, we can simply introduce an event function to detect when the ball hits the ground, *i.e.* $g(t, z(t), \phi) = x(t) - r$. We can then instantaneously modify the state so that $z'(t^*) = [x(t^*), -(1 - \alpha)v(t^*)]$, where $\alpha$ is the fraction of momentum that is absorbed by the contact, and then resume solving the ODE in eq. (2) with this new state $z'(t^*)$.

Figure 1 shows the bouncing ball example being fit by a Neural ODE and a Neural *Event* ODE where both $f$ and $g$ are neural networks. The Neural ODE model parameterizes a non-linear function for $f$ while the Neural Event ODE parameterizes $f$ and $g$ as linear functions of $z(t)$. We see that the Neural Event ODE can perfectly recover the underlying physics and extrapolate seamlessly. Meanwhile, the Neural ODE has trouble fitting to the sudden changes in dynamics when the ball bounces off the ground, and furthermore, does not generalize because the true model requires the trajectory to be discontinuous.

The Neural Event ODE, while being capable of modeling discontinuities in $t$, is a continuous function of the parameters and hence can be trained with gradient descent. Going forwards, we will discuss how to differentiate $t^*$ w.r.t. the variables that depend on it, such as $z(t^*)$, $\phi$ and $\theta$. Before this, we briefly summarize how gradients can be computed through any black-box ODE solver.

## 2 BACKGROUND: DIFFERENTIATING THROUGH ODE SOLUTIONS

Consider a scalar-valued loss function $L$ that depends on the output of an ODE solver,

$$L(z(\tau)) = L(\texttt{ODESolve}(z_0, f, t_0, \tau, \theta)) \tag{4}$$

where $f(t, z, \theta)$ describes the dynamics. To optimize $L$, we require the gradients with respect to each of the inputs: $z_0$, $t_0$, $\tau$ and $\theta$. All of these inputs influence the loss function through the intermediate states $z(t)$, for $t \in [t_0, \tau]$, and their gradients can be expressed in relation to the *adjoint* state $a(t) \triangleq \frac{dL}{dz(t)}$ which contains the gradient of all intermediate states.

The adjoint method (see *e.g.* Pontryagin et al., 1962; Le Cun, 1988; Giles & Pierce, 2000; Chen et al., 2018) provides an identity that quantifies the instantaneous change in the adjoint state:

$$\frac{da(t)}{dt} = -a(t)^\mathsf{T} \frac{\partial f(t, z(t), \theta)}{\partial z} \tag{5}$$

which when combined with $z(t)$ is an ordinary differential equation that—by solving the adjoint state backwards-in-time, similar to a continuous-time chain rule—allows us to compute *vector-Jacobian products* of the form $v^\mathsf{T} \left[ \frac{\partial z(\tau)}{\partial \xi} \right]$, where $\xi$ is any of the inputs $z_0, t_0, \tau, \theta$. For instance, with $v = \frac{dL}{dz(\tau)}$, the product $\frac{dL}{dz_0} = \frac{dL}{dz(\tau)}^\mathsf{T} \frac{dz(\tau)}{dz_0}$ effectively propagates the gradient vector from the final state, $\frac{dL}{dz(\tau)}$, to the intial state, $\frac{dL}{dz_0}$. The ability to propagate gradients allows $\texttt{ODESolve}$ to be used within reverse-mode automatic differentiation (Baydin et al., 2018).

We use the method in Chen et al. (2018), which solves the adjoint state and parameter gradients jointly backwards-in-time alongside the state $z(t)$. This method does not require intermediate values of $z(t)$ to be stored and only invokes $\texttt{ODESolve}$ once for gradient computation.

There exist other notable approaches for solving the adjoint equations with different memory-compute trade-offs, such as storing all intermediate quantities (also known as discrete adjoint) (*e.g.* Zhang & Sandu 2014), more sophisciated methods of checkpointing (Chen et al., 2016; Gholami et al., 2019; Zhuang et al., 2020), the use of interpolation schemes (Hindmarsh et al., 2005; Daulbaev et al., 2020), and symplectic integration (Zhuang et al., 2021; Matsubara et al., 2021). Any of these approaches can be used and is tangential to our contributions.

## 3 DIFFERENTIATING THROUGH EVENT HANDLING

In an event-terminated ODE solve, the final time value $t^*$ is not an input argument but a function of the other inputs. As such, for gradient-based optimization, we would need to propagate gradients from $t^*$ to the input arguments of $\texttt{ODESolveEvent}$ (eq. (3)).

Consider a loss function $L$ that depends on the outputs of $\texttt{ODESolveEvent}$,

$$L(t^*, z(t^*)) \qquad \text{where } t^*, z(t^*) = \texttt{ODESolveEvent}(z_0, f, g, t_0, \theta, \phi). \tag{6}$$

Without loss of generality, we can move the parameters $\phi$ inside the state $z_0$ and set $\frac{d\phi(t)}{dt} = 0$. As long as we can compute gradients w.r.t $z_0$, these will include gradients w.r.t. $\phi$. This simplifies the event function to $g(t, z)$.

Furthermore, we can interpret the event function to be solving an ODE at every evaluation (as opposed to passing the event function as an input to an ODE solver) conditioned on the input arguments. This simplifies the event handling procedure to finding the root of

$$g_{\text{root}}(t, z_0, t_0, \theta) \triangleq g\big(t, z = \texttt{ODESolve}(z_0, f, t_0, t, \theta)\big) \tag{7}$$

and factorizes the `ODESolveEvent` procedure into two steps:

$$\begin{cases} t^* & = \arg\min_t \ t \geq t_0 \text{ such that } g_{\text{root}}(t, z_0, t_0, \theta) = 0 \\ z(t^*) & = \texttt{ODESolve}(z_0, f, t_0, t^*, \theta). \end{cases} \tag{8}$$

It is obviously computationally infeasible to numerically solve an ODE within the inner loop of a root finding procedure, but this re-interpretation allows us to use existing tools to derive the gradients for `ODESolveEvent` which can be simplified later to just solving one ODE backwards-in-time.

First, the implicit function theorem (Krantz & Parks, 2012) gives us the derivative from $t^*$ through the root finding procedure. Let $\xi$ denote any of the inputs $(z_0, t_0, \theta)$. Then the gradient satisfies

$$\frac{dt^*}{d\xi} = -\left(\frac{dg_{\text{root}}(t^*, z_0, t_0, \theta)}{dt}\right)^{-1}\left[\frac{\partial g(t^*, z(t^*))}{\partial z}\frac{\partial z(t^*)}{\partial \xi}\right]. \tag{9}$$

Though $g_{\text{root}}$ requires solving an ODE, the derivative of $z(t^*)$ w.r.t. $t^*$ is just $f^* \triangleq f(t^*, z(t^*))$, so

$$\frac{dg_{\text{root}}(t^*, z_0, t_0, \theta)}{dt} = \frac{\partial g(t^*, z(t^*))}{\partial t} + \frac{\partial g(t^*, z(t^*))}{\partial z}^{\mathsf{T}} f^*. \tag{10}$$

Taking into account that the loss function may directly depend on both $t^*$ and $z(t^*)$, the gradient from the loss function $L$ w.r.t. an input $\xi$ is

$$\frac{dL}{d\xi} = \underbrace{\left(\frac{\partial L}{\partial t^*} + \frac{\partial L}{\partial z(t^*)}^{\mathsf{T}} f^*\right)}_{\frac{dL}{dt^*}} \underbrace{\left(-\frac{dg_{\text{root}}(t^*, z_0, t_0, \theta)}{dt}^{-1}\left[\frac{\partial g(t^*, z(t^*))}{\partial z}^{\mathsf{T}}\frac{\partial z(t^*)}{\partial \xi}\right]\right)}_{\frac{dt^*}{d\xi}} + \frac{\partial L}{\partial z(t^*)}^{\mathsf{T}}\left[\frac{\partial z(t^*)}{\partial \xi}\right]. \tag{11}$$

Re-organizing this equation, we can reduce this to

$$\frac{dL}{d\xi} = v^{\mathsf{T}}\left[\frac{\partial z(t^*)}{\partial \xi}\right] \tag{12}$$

where $v = \left(\frac{\partial L}{\partial t^*} + \frac{\partial L}{\partial z(t^*)}^{\mathsf{T}} f^*\right)\left(-\frac{dg_{\text{root}}(t^*, z_0, t_0, \theta)}{dt}^{-1}\frac{\partial g(t^*, z(t^*))}{\partial z}\right) + \frac{\partial L}{\partial z(t^*)}$.

All quantities in $v$ can be computed efficiently since $g$ and $t^*$ are scalar quantities. As they only require gradients from $g$, there is no need to differentiate through the ODE simulation to compute $v$. Finally, the vector-Jacobian product in eq. (12) can be computed with a single `ODESolve`.

We implemented our method in the `torchdiffeq` (Chen, 2018) library written in the Py-Torch (Paszke et al., 2019a) framework, allowing us to make use of GPU-enabled ODE solvers. We implemented event handling capabilities for all ODE solvers in the library along with gradient computation for event functions.

Differentiable event handling generalizes many numerical methods that often have specialized methods for gradient computation, such as ray tracing (Li et al., 2018), physics engines (de Avila Belbute-Peres et al., 2018; Hu et al., 2020), and spiking neural networks (Wunderlich & Pehle, 2020).

### 3.1 NEURAL EVENT FUNCTIONS AND INSTANTANEOUS UPDATES

This allows the use of `ODESolveEvent` as a differentiable modeling primitive, available for usage in general deep learning or reverse-mode automatic differentiation libraries. We can then construct a Neural Event ODE model that is capable of modeling a variable number of state discontinuities, by repeatedly invoking `ODESolveEvent`. The Neural Event ODE is parameterized by a drift function, an event function, and an instantaneous update function that determines how the event is updated after each event. All functions can have learnable parameters. The model is described concretely in Algorithm 1 which outputs event times and the piecewise continuous trajectory $z(t)$.

## 4 GENERAL STATE-DEPENDENT EVENT FUNCTIONS

The state-dependent event function is the most general form and is suitable for modeling systems that have discontinuities when crossing boundaries in the state space. We apply this to modeling switching dynamical system and physical collisions in a multi-body system.

---

**Algorithm 1** The Neural Event ODE. In addition to an ODE, we also model a variable number of possible event locations and how each event affects the system.

---

**Input:** An initial condition $(t_0, z_0)$, a final termination time $T$, a drift function $f$, an event function $g$, and an instantaneous update function $h$. *Parameters of $f, h, g$ are notationally omitted.*
$i = 0$
**while** $t_i < T$ **do**
$\quad t_{i+1}, z'_{i+1} = \text{ODESolveEvent}(z_i, f, g, t_i)$ $\qquad\qquad$ ▷ Solve until the next event
$\quad z_{i+1} = h(t_{i+1}, z'_{t+1})$ $\qquad\qquad\qquad$ ▷ Instantaneously update the state
$\quad i = i + 1$
**end while**
**Return:** event times $\{t_i\}$ and the piecewise continuous trajectory $\{z_i(t)$ for $t_i \leq t \leq t_{i+1}\}$

---

## 4.1 SWITCHING LINEAR DYNAMICAL SYSTEMS

Switching linear dynamical systems (SLDS) are hybrid discrete and continuous systems that choose which dynamics to follow based on a discrete switch (see *e.g.* Ackerson & Fu 1970; Chang & Athans 1978; Fox et al. 2009; Linderman et al. 2016). They are of particular interest for extracting interpretable models from complex dynamics in areas such as neuroscience (Linderman et al., 2016) or finance (Fox et al., 2009).

Here, we consider linear switching dynamical systems in the continuous-time setting, placing them in the framework of event handling, and evaluate whether a Neural Event ODE can recover the underlying SLDS dynamics. At any time, the state consists of a continuous variable $z$ and a switch variable $w$. Representing $w$ as a one-hot vector, the drift function is then $\frac{dz(t)}{dt} = \sum_{m=1}^{M} w_m \left( A^{(m)} z + b^{(m)} \right)$, where $M$ is the number of switch states. The switch variable $w$ can change instantaneously, resulting in discontinuous dynamics at the moment of switching.

**Setup** We adapt the linear switching dynamical system example of a particle moving around a synthetic race track from Linderman et al. (2016) to continuous-time and use a fan-shaped track. We created a data set with 100 short trajectories for training and 25 longer trajectories as validation and test sets. This was done by sampling a trajectory from the ground-truth system and adding random noise. Figure 2 shows the ground-truth dynamics and sample trajectories. Given observed samples of trajectories from the system we learn to recover the event and state updates by minimizing the squared error between the predicted and actual trajectories. Further details are in App. A.

**Results** For our Neural Event ODE, we relax the switch state to be a positive real vector that sums to one. At event locations, we instantaneously change the switch state $w$ based on the event location. We evaluated our model in comparison to a non-linear Neural ODE that doesn't use event handling and a recurrent neural network (RNN) baseline. We show the test loss in table 1, and the visualization in fig. 2 demonstrates that we are able to recover the components and can generate realistic trajectories with sharp transitions. We note that since the switch variable is only set at event locations, they only need to be accurate at the event locations.

Table 1: Continuous-time switching linear dynamical system.

| Model | Test Loss (MSE) |
|---|---|
| RNN (LSTM) | $0.261 \pm 0.078$ |
| Neural ODE | $0.157 \pm 0.005$ |
| Neural Event ODE | $0.093 \pm 0.037$ |

## 4.2 MODELING PHYSICAL SYSTEMS WITH COLLISION

We next consider the use of event handling for modeling physical collisions. Though there exist physics-motivated parameterizations of Neural ODEs (*e.g.* Zhong et al., 2019; Greydanus et al., 2019; Finzi et al., 2020), these cannot model collision effects. Event handling allows us to capture discontinuities in the state (from collisions) that otherwise would be difficult and unnatural to model with a Neural ODE alone. Building on the bouncing ball example from sect. 1.1, we consider a multi-body 2D physics setting with collisions.

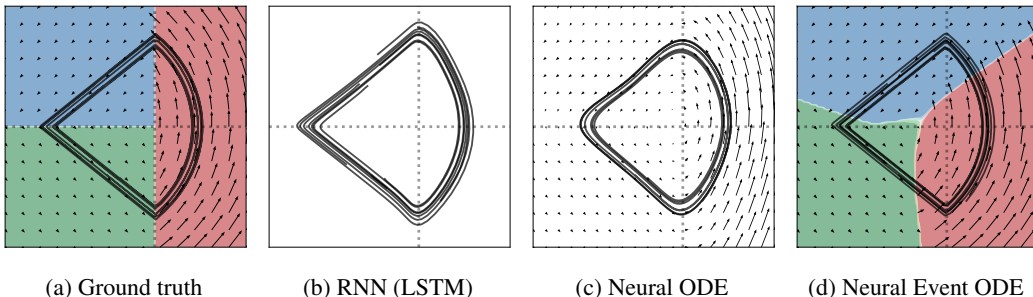

| (a) Ground truth | (b) RNN (LSTM) | (c) Neural ODE | (d) Neural Event ODE |

Figure 2: We learn continuous switching linear dynamical systems with Neural Event ODE to model and predict the trajectory of a particle traveling on a loop with discontinuous changes in dynamics.

Table 2: Test results on the physics prediction task. $@N$ notates evaluating on a sequence length of $N$ discretized steps. As a proxy for the complexity of ODE models, we also report the number of function evaluations (NFE) for the ODE dynamics and the event function.

| | Test Loss (MSE) | | | Complexity (NFE) ($\times 10^3$) | | | |
|---|---|---|---|---|---|---|---|
| Model | @25 | @50 | @100 | ODE@50 | ODE@100 | EventFn@50 | EventFn@100 |
| RNN (LSTM) | $0.01 \pm 0.00$ | $0.07 \pm 0.03$ | $0.24 \pm 0.02$ | — | — | — | — |
| Neural ODE | $0.00 \pm 0.00$ | $0.06 \pm 0.01$ | $0.18 \pm 0.02$ | $3.81 \pm 0.24$ | $7.82 \pm 0.43$ | — | — |
| Neural Event ODE | $0.01 \pm 0.00$ | $0.07 \pm 0.00$ | $0.19 \pm 0.02$ | $0.16 \pm 0.00$ | $0.35 \pm 0.01$ | $0.16 \pm 0.01$ | $0.36 \pm 0.01$ |

**Setup** We use Pymunk/Chipmunk (Blomqvist, 2011; Lembcke, 2007) to create a data set of trajectories from simulating two balls colliding in a box with random initial positions and velocities. As part of our model, we learn (i) a neural event function to detect collision either between the two balls or between each ball and the ground; (ii) an instantaneous update to the state to reflect a change in direction due to contact forces. Both the event function and instantaneous updates are parameterized as deep neural networks. Further details and exact hyperparameters are described in App. A.

**Results** We evaluate our model in comparison to a RNN baseline and a non-linear Neural ODE that doesn't use events. The non-linear Neural ODE is parameterized such that the velocity is the change in position, but the change in velocity is a non-linear neural network. We report the test loss in table 2, and the complexity of the learned dynamics. The Neural Event ODE generalizes better than RNN baseline and matches the non-linear Neural ODE. Though the non-linear Neural ODE can perform very well, it must use a very stiff dynamics function to model sudden changes in velocity at collisions. In contrast, the Neural Event ODE requires much fewer function evaluations to solve and can output predictions much faster.

We illustrate some sample trajectories in fig. 3. We find that the RNN and Neural ODE baselines learn to reduce the MSE loss of short bounces by simply hovering the ball in mid-air. On the other hand, the Neural Event ODE can exhibit difficulty recovering from early mistakes which can lead to chaotic behavior for longer sequences.

## 5 THRESHOLD-BASED EVENT FUNCTIONS

We next discuss a special parameterization of event functions that are based on an integrate-and-threshold approach. This event function is coupled with the state and depends on an accumulated integral. That is, let $\lambda(t) \in \mathbb{R}^+ \cup \{0\}$ be a positive quantity. Then a threshold-based event occurs when the integral over $\lambda$ reaches some threshold $s$. The event time is thus the solution to

$$t^* \text{ such that } s = \int_{t_0}^{t^*} \lambda(t) \, dt \tag{13}$$

which can be implemented with an `ODESolveEvent` by tracking $\Lambda(t) \triangleq \int_{t_0}^{t} \lambda(s) ds$ as part of the state $z(t)$ and using the event function $g(t, z(t)) = s - \Lambda(t)$.

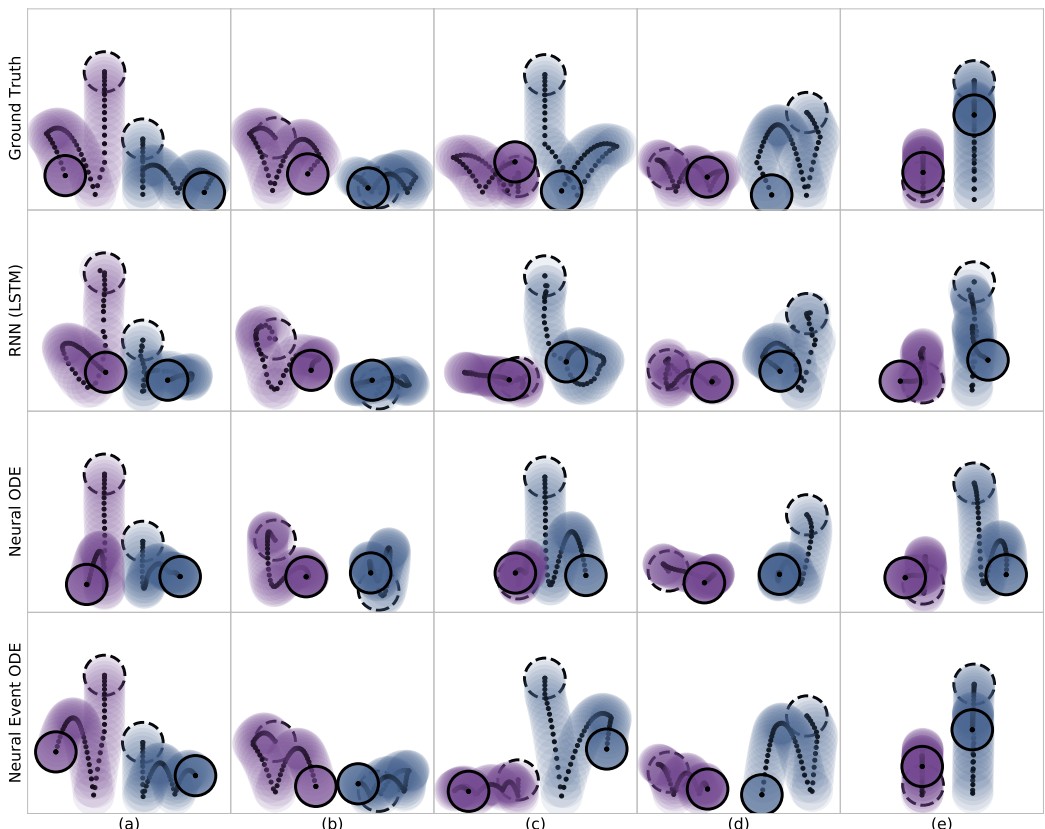

Figure 3: Test-set motion predictions of two bouncing balls with collisions. The dashed black circle indicates the initial positions of the balls and the solid black circle indicates the final positions. The Neural Event ODE model is able to recover realistic event and state updates that exhibit floating behavior (a-c) and straighter paths (d-e) than a RNN baseline. (c) shows a failure mode of the RNN and Neural Event ODE models where the first collision is not correctly predicted.

This form appears in multiple areas such as neuronal dynamics (Abbott, 1999; Hodgkin & Huxley, 1952), inverse sampling (Steinbrecher & Shaw, 2008) and more generally temporal point processes. We focus our discussion around temporal point processes as they encompass other applications.

**Temporal point processes (TPPs)**    The TPP framework is designed for modeling random sequences of event times. Let $\mathcal{H} = \{t_i\}_{i=1}^n$ be a sequence of event times, with $t_i \in \mathbb{R}$ and $i \in \mathbb{Z}^+$. Additionally, let $\mathcal{H}(t) = \{t_i \mid t_i < t, t_i \in \mathcal{H}\}$, *i.e.* the history of events predating time $t$. A temporal point process is then fully characterized by a *conditional intensity function* $\lambda^*(t) = \lambda(t \mid \mathcal{H}(t))$. The star superscript is a common shorthand used to denote conditional dependence on the history (Daley & Vere-Jones, 2003). The only condition is that $\lambda^*(t) > 0$. The joint log likelihood of observing $\mathcal{H}$ starting with an initial time value at $t_0$ is

$$\log p\left(\{t_i\}\right) = \sum_{i=1}^n \log \lambda^*(t_i) - \int_{t_0}^{t_n} \lambda^*(\tau)\,d\tau. \tag{14}$$

In the context of flexible TPP models parameterized with neural networks, Mei & Eisner (2017) used a Monte Carlo estimate of the integral in eq. (14), Omi et al. (2019) directly parameterized the integral instead of the intensity function, and Jia & Benson (2019) noted that this integral can be computed using an ODE solver. While these approaches can enable training flexible TPP models by maximizing log-likelihood, it is much less straightforward to learn from simulations.

In the following, we discuss how the event function framework allows us to backpropagate through simulations of TPPs. This enables training TPPs with the "reverse KL" objective. Another form

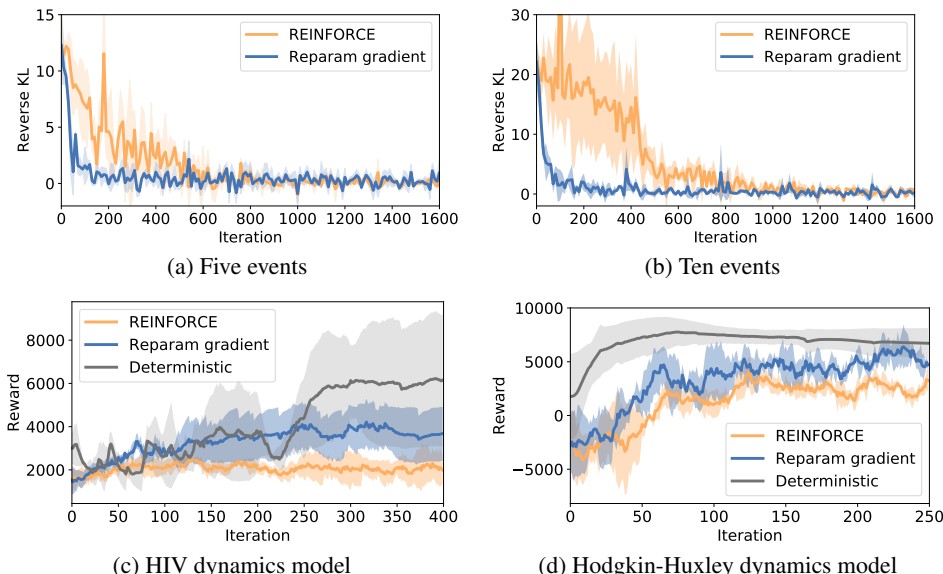

Figure 4: Learning TPPs (a,b) through reverse KL and (c,d) for discrete control in continuous time. Our reparameterization gradient improves upon the REINFORCE gradient and we can even learn a *deterministic* discrete-valued policy.

of simulation-based training appears in reinforcement learning, where a TPP policy can be used to perform instantaneous interventions on a continuous-time environment. Our method can also be readily applied to extensions such as spatio-temporal point processes (Chen et al., 2021).

## 5.1 REPARAMETERIZATION GRADIENT FOR TEMPORAL POINT PROCESSES

Sampling a single event can be done in two steps: (i) sample $s \sim \text{Exp}(1)$ and (ii) solve for $t^*$ such that $s = \int_0^{t^*} \lambda^*(t)dt$, which is exactly of the form eq. (13). This allows us to reparameterize the sample $t^*$ as a transformation of a noise variable $s$, thus allows us to take gradients of samples with respect to parameters of the TPP. Consider a conditional intensity function $\lambda_\theta^*$ parameterized by $\theta$,

$$\nabla_\theta \mathbb{E}_{\{t_i\} \sim \text{TPP}(\lambda_\theta^*)} \left[ f(\{t_i\}) \right] = \mathbb{E}_{\{s_i\} \sim \text{Exp}(1)} \left[ \nabla_\theta f(\{t_i(s_i, \theta)\}) \right] \tag{15}$$

provides the reparameterization gradient, where $t_i(s_i, \theta)$ is the solution to step (ii) above, and can be implemented as an `ODESolveEvent` which is differentiable.

We compare the reparameterization gradient against the REINFORCE gradient (Sutton et al., 2000) in training a Neural Jump SDE (Jia & Benson, 2019) with a reverse KL objective

$$\mathcal{D}_{\text{KL}}(p_\theta, p_{\text{target}}) = \mathbb{E}_{\{t_i\} \sim \text{TPP}(\lambda_\theta)} \left[ \log p_\theta(\{t_i\}) - \log p_{\text{target}}(\{t_i\}) \right] \tag{16}$$

where $p_{\text{target}}$ is taken to be a Hawkes point process. The reparameterization gradient of this objective requires simulating from the model and requires taking gradients through the event handling procedure. Results are shown in figs. 4a and 4b. The REINFORCE gradient is generally perceived to have a higher variance and slower convergence, which is also reflected in our experiments. The reparameterization gradient performs well on sequences of either five or ten events, while the REINFORCE gradient exhibits slower convergence with longer event sequences.

**Learning discrete control over continuous-time systems** We show that the reparameterization gradient also allows us to model discrete-valued control variables in a continuous-time system using Neural Jump SDEs as control policies. We use a multivariate TPP, where a sample from a particular dimension, or index, changes the control variable to a discrete value corresponding to that index.

We test using the human immunodeficiency virus (HIV) dynamics model from Adams et al. (2004) which simulates interventions from treatment strategies with inhibitors. We use a discrete control

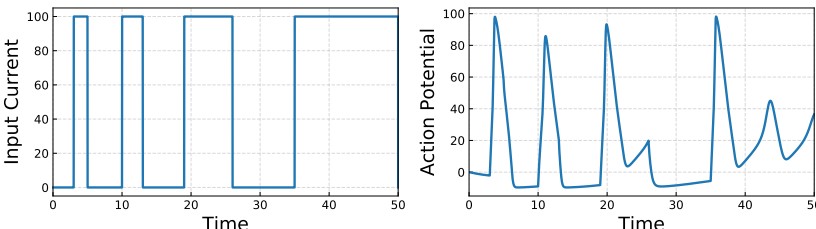

Figure 5: Visualization of a discrete-valued control (*left*) in a continuous-time environment (*right*), using the Hodgkin-Huxley model of neuronal dynamics.

model for determining whether each of two types of inhibitors should be used, resulting in 4 discrete states, similar to the setup in Miller et al. (2020). The reward function we use is the same as that of Adams et al. (2004). We additionally experiment with the Hodgkin-Huxley neuronal dynamics model (Hodgkin & Huxley, 1952), where we have an input stimulus that can switch between being on or off at any given time. The reward is high if the action potentials from the Hodgkin-Huxley model match that of a target action potential. See fig. 5. All experiments were run with three seeds. Detailed setup in App. A.

Results are shown in figs. 4c and 4d, which shows that the reparameterization gradient outperforms REINFORCE in both settings. Interestingly, we can also train with a deterministic control policy: instead of randomly sampling $s_i$, we fix these threshold values. This deterministic policy can outperform both stochastic policies as the underlying continuous-time system is deterministic.

## 6   SCOPE AND LIMITATIONS

**Minibatching**   Each trajectory in a minibatch can have different event locations. However, our ODE solvers do not have an explicit notation of independence between systems in a minibatch. For batching, we can combine a minibatch of ODEs into one system and then use a sparse aggregator (*e.g.* min or max operator) on the event functions. This does introduce some overhead as we would restart integration for the combined system whenever one event function in the minibatch triggers.

**Runaway event boundaries**   In some cases, a neural event function is never triggered. In such cases, there will be no gradient for the event function parameters and the model degenerates to a Neural ODE alone. In high dimensions, the roots of a neural network are in general unpredictable and a trajectory may never encounter a root. We have found that initializing the parameters of the event function with a large standard deviation helps alleviate this problem for general neural event functions. This is less of a problem for the threshold-based event functions, as the integrand is always positive and there will always be an event if the state is solved for long enough.

**Numbers of events is discrete**   An objective function that relies on the number of events can be discontinuous with respect to parameters of the event function, as a small change in parameter space can introduce a jump in the number of events. Ultimately, this is a matter of the formulation in which gradients are used. For instance, in the stochastic TPP setting, the reparameterization gradient relies on the function being differentiable (Mohamed et al., 2019). For this reason, Shchur et al. (2020) used surrogate objectives for sampling-based training.

## 7   CONCLUSION

We consider parametrizing event functions with neural networks in the context of solving ODEs, extending Neural ODEs to implicitly defined termination times. This enables modeling discrete events in continuous-time systems—*e.g.* the criteria and effects of collision in physical systems—and simulation-based training of temporal point processes with applications to discrete control. Notably, we can train deterministic policies with discrete actions, by making use of gradients that only exist in a continuous-time setting.

ACKNOWLEDGEMENTS

We thank David Duvenaud for helpful discussions. Additionally, we acknowledge the Python community (Van Rossum & Drake Jr, 1995; Oliphant, 2007) for developing the core set of tools that enabled this work, including PyTorch (Paszke et al., 2019b), torchdiffeq (Chen, 2018), higher (Grefenstette et al., 2019), Hydra (Yadan, 2019), Jupyter (Kluyver et al., 2016), Matplotlib (Hunter, 2007), seaborn (Waskom et al., 2018), numpy (Oliphant, 2006; Van Der Walt et al., 2011), pandas (McKinney, 2012), and SciPy (Jones et al., 2014).

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

## A  EXPERIMENTAL DETAILS

**Continuous-time Switching Linear Dynamical Systems**

DATA  We constructed a fan-shaped system, similar to a baseball track. Let $A$ be the rotation matrix

$$\begin{bmatrix} 0 & 1 \\ -1 & 0 \end{bmatrix} \tag{17}$$

Then the ground truth dynamical system followed

$$\frac{dx}{dt} = \begin{cases} xA + \begin{bmatrix} 0 & 2 \end{bmatrix} & \text{if } x_1 \geq 2 \\ \begin{bmatrix} -1 & -1 \end{bmatrix} & \text{if } x_0 \geq 0 \text{ and } x_1 < 2 \\ \begin{bmatrix} 1 & -1 \end{bmatrix} & \text{if } x_0 < 0 \text{ and } x_1 < 2 \end{cases} . \tag{18}$$

We discretized 100 sequences of length 4 into 50 discrete steps for training. For each of validation and test, we discretized 25 sequences of length 12 into 150 discrete steps. We added independent Gaussian noise with standard deviation 0.05 to training sequences.

ARCHITECTURE  We set our dynamics function to be a weighted form of eq. (18), where the weights $w$ did not change over time and were only modified at event times. We parameterize the event function to be the product of two tanh-gated outputs of a linear function.

$$\text{EventFn}(x) = \prod_i [\tanh(Wx + b)]_i \tag{19}$$

And we parameterized the instantaneous change to weights as a neural network with 2 hidden layers, each with 1024 hidden units, with ReLU activation functions. The output is passed through a softmax to ensure the weights sum to one.

BASELINES  We modeled the drift for the non-linear Neural ODE baseline as a MLP with 2 hidden layers with 256 hidden units each and the ReLU activation function. The LSTM baseline has a hidden state size of 128, and the hidden state is passed through a MLP decoder (with 2 hidden layers of 128 units each and the ReLU activation) to predict the displacement in position between each time step. LSTMs that directly predicts the position was also tested, but could not generalize well.

TRAINING  We use a mean squared loss on both the position and the change in position. Let $x_t$, $t = 1, \ldots, N$, be the ground truth positions and $\widehat{x}_t$ the model's predicted positions. The training objective is then

$$\left[ \frac{1}{N} \sum_{t=1}^{N} (x_t - \widehat{x}_t)^2 \right] + \lambda \left[ \frac{1}{N-1} \sum_{t=1}^{N-1} ((x_{t+1} - x_t) - (\widehat{x}_{t+1} - \widehat{x}_t))^2 \right] \tag{20}$$

where we tried $\lambda \in \{0, 0.01\}$ and chose $\lambda = 0.01$ for all models as the validation was lower. For validation and test, we only used the mean squared error on the positions $x_t$. For optimization, we used Adam with the default learning rate of 0.001 and a cosine learning decay. All models were trained with a batch size of 1 for 25000 iterations.

**Modeling Physical Systems with Collision**

DATA  We used the Pymunk/Chipmunk (Blomqvist, 2011; Lembcke, 2007) library to simulate two balls of radius 0.5 in a $[0, 5]^2$ box. The initial position is randomly sampled and the initial velocity is zero. We then simulated for 100 steps. We sampled 1000 initial positions for training and 25 initial positions each for validation and test.

ARCHITECTURE  We parameterized the event function as a deep neural network. The outputs are then passed through a tanh and then multiplied together to form a single scalar. The event function took as input the positions of the two balls.

$$\text{EventFn}(x) = \prod_{i=1}^{M} [\tanh(\text{MLP}(x))]_i \tag{21}$$

where $M = 8$. The neural network is a multilayer perceptron (MLP) with 1 hidden layer of 128 hidden units. We parameterized the instantaneous update function to be a MLP with 3 hidden layers

with 512 hidden units. The instantaneous update took as input the position, velocity, and the pre-tanh outputs of the neural event function. We used relative and absolute tolerances of 1E-8 for solving the ODE.

BASELINES The recurrent neural network baseline uses the LSTM architecture (Hochreiter & Schmidhuber, 1997) and outputs the difference in position between time steps, which we found to be more stable than directly predicting the absolute position. The non-linear Neural ODE baseline uses the domain knowledge that velocity is the change in position. We then used a MLP with 2 hidden layers and 256 hidden units to model the instantaneous change in velocity.

TRAINING We used a mean squared loss on the position of the two balls. For optimization, we used Adam with learning rate 0.0005 for the event function and 0.0001 for the instantaneous update. We also clipped gradient norms at 5.0. All models were trained for 1000000 iterations, where each iteration used a subsequence of 25 steps as the target.

**Discrete Control over Continuous-time Systems**

HIV DYNAMICS We used the human immunodeficiency virus (HIV) model of Adams et al. (2004), which describes the dynamics of infected cells depending on treatments representing reverse transcriptase (RT) in hibitors and protease inhibitors (PIs). We used a discrete-valued treatment control as was done by Miller et al. (2020) where RT could either be used (with a strength of 0.7) or unused, and PI could either be used (with a strength of 0.3) or unused. Thus this results in a discrete control variable with $k = 4$ states. The reward is the same as in Adams et al. (2004).

HODGKIN-HUXLEY NEURONAL DYNAMICS This model describes the propagation of action potentials in neurons (Hodgkin & Huxley, 1952; Gerstner et al., 2014). We use a discrete valued input current as the control variable, with values 0 or 100. This then stimulates an action potential following the Hodgkin-Huxley model. We use a specfic action potential pattern as the target $A^*$, and train a control policy that can recover this action potential. The reward is set to an integral over the infinitesimal negative squared error at each time value.

$$\text{Reward} = \int_{t_0}^{t_1} \left( -(A(t) - A^*(t))^2 - b \right) \ dt \tag{22}$$

where $b = 600$ acts a baseline for the REINFORCE gradient estimator. Figure 5 shows the target action potential $A^*$ and the input current used to generate it.

CONTROL / POLICY We used a multivariate temporal point process (TPP) with $k$ dimensions where a sample from each dimension instantaneously changes the control variable to the discrete state corresponding to that dimension. The conditional intensity function of the TPP was parameterized as a function of a continous-time hidden state of width 64. The continuous-time hidden state followed a neural ODE with 3 hidden layers and sine activation functions. The hidden state is instantaneously modified by a MLP with 2 hidden layers and sine activation functions. The conditional intensity function is a function of the hidden state and is modeled by a MLP with 1 hidden layer and ReLU activation. All hidden dimensions were 64.

TRAINING All models were trained using Adam with learning rate 0.0003.

