# OpenReview forum: "Learning Neural Event Functions for Ordinary Differential Equations"
_ICLR.cc/2021/Conference — ICLR 2021 Poster_

### Official Review · AnonReviewer4 · 2020-10-28
**Neural event functions for ODEs**

**Rating:** 6
**Confidence:** 4

**Review:**

Summary
--

This authors extend neural ODEs to implicitly defined termination criteria modelled by 'neural event functions'. This allows neural ODEs to model abrupt changes in the dynamics (such as collisions or switching dynamics). The authors present how the even handling can be differentiated through, and include a representative set of example studies in the experiments.

The paper is well written, easy to follow, and presents relevant background and related work. The idea extends the already well-studied methodology related t neural ODE models, and even if the idea behind the contribution is small, the paper should be of interest for the audience of ICLR. The paper reads well, and the examples and illustrations help understanding.

I liked this paper and found it interesting and well presented. The concerns below are reflected in my score.


Concerns
--

1. The main reason for my low positive score is that the originality of the idea is rather low, and I view the paper mostly as an incremental addition to the toolbox of neural ODE methods. Nevertheless, I found the paper interesting and worthy of publication. The well-chosen experiments and illustrative examples further help communicate the idea, which I see as a strong point of this paper.

2. In the experiments, it would have been interesting to see comparisons to not purely learning based models. This applies especially to the collision example, where a more classical method (which would most likely require more hand-tailored prior knowledge of the task) would have been a good baseline (showing how close or well these models get to a customised baseline model). I am fully aware that this would require a considerable amount of work and these comparisons are often omitted in ML papers. Maybe something to think of still.

3. Minor: I recommend going through the paper and making sure especially citations and equations read well as part of the main text.

---

> ### Author Response · Authors · 2020-11-19
> **Thank you for your review and suggestions**
>
> We thank the reviewer for the direct review. We agree that part of our technical contribution consists of applying the implicit function theorem is a perhaps simple-in-hindsight re-interpretation of event handling in ODE solvers. Though we do believe our contributions as a whole are much more meaningful than an incremental addition.
>
> Previously event functions were human designed, heavily domain-dependent, and whether learning them is even possible was not explored. Furthermore, we showed a preview of the potential application areas where having differentiable event functions are beneficial. Thus we’ve shown both that training event functions is viable and have motivated their usefulness.
>
> While some dynamics can be modeled with a neural ODE without events, the learned ODE becomes very stiff around the regions where a discontinuity is more accurate. As a result, an ODE-only model requires substantially more computation to simulate than using events (as we quantitatively show in the updated Table 2). The ability to incorporate constraints and boundaries into our models is also useful, for instance, in modeling in robotics, where joints have complex constraints that must be satisfied.
>
> We derive and implement the reparameterization gradient for intensity-based temporal point processes, which while relatively straightforward in hindsight, has not been discussed in the literature. We’re also very excited about the ability to model deterministic discrete control variables in continuous-time, whereas previous works could only train using stochastic gradients from a probabilistic model (either SLDS or TPP). We see many potential applications where discrete actions are performed in a continuous-time environment that can benefit from our approach, such as modeling chemical or nuclear reactions, motion planning, discrete control, event-based sampling, inference in systems with jumps, etc.
>
> 2. We thank the reviewer for this suggestion. We couldn’t think of a suitable comparison at this moment. A gradient-free optimization method could make a good comparison; but methods such as Bayesian optimization or evolutionary strategy likely won’t work very well in learning neural event functions that have high dimensional parameter spaces, and on the other hand, may simply become a row of zeros in our table if we gave too much domain knowledge for the applications we explored (e.g. only learning the size of the balls or the gravity coefficient of the physics simulator), which also becomes an uninformative comparison. We do note that the REINFORCE gradient used for comparison in the point process experiments is similar to a sophisticated random search, as it does not make use of gradient information from the objective. It is also a current state-of-the-art approach for stochastic control using TPPs [1] Going forward, we will definitely keep this suggestion in mind for future applications/domains.
>
> 3. Good catch; we have updated the coherence of some equations and citations.
>
> [1] Deep Reinforcement Learning of Marked Temporal Point Processes; Utkarsh Upadhyay, Abir De, and Manuel Gomez-Rodriguez; 2018

---

### Official Review · AnonReviewer2 · 2020-10-28
**Promising research direction, lacks stronger motivation and experimental evaluation**

**Rating:** 6
**Confidence:** 3

**Review:**

Summary: The paper provides a follow up approach to the Neural ODE [Chen et al., 2019] approach, where the termination time is implicitly defined by neural event functions rather than being pre-determined -- which is a promising direction for dealing with state discontinuities. The proposed method is tested on simple dynamical and two-body systems.

Strengths:
 + Paper is well written
 + Promising direction for dealing with state discontinuities
 + Code will be released upon publication

Weak Points:

1) IMO, an explicit algorithm description should be included in the manuscript.

2) I found the experimental part of the paper to be weak. Overall lots of model details are missing, comparison to baselines should include extra analysis and the approach should be tested on more challenging domains. More specifically,

a) When comparing to LSTMs and Neural ODEs, only sample trajectories and test MSE loss are presented. A precise description of the comparison baselines is missing (number of layers, hidden sizes, optimization setup, etc). An analysis comparing the number of parameters of each baseline would also be interesting. What is the odesolve used? Lots of missing details.

b) How does the number of iterations on the ODE solve behave wtr to the errors? Do we see a similar behavior as in Neural ODE, where the number of function evaluations that the ODE solver makes increases along training?

c) in the collision example, table 2 indicates neural ODE has similar (if not better) performance than the proposed approach. What happens when you have more bouncing balls?

d) In the temporal point process examples from Figure 4, I think a more detailed problem overview (even if on the appendix) would help to understand the relevance of the results.

e) Have you tried this approach on normalizing flows?

f) In Figure 2, authors should explain the color code used.

So although I like the general research direction this paper takes, I think the experimental setup needs to be extended significantly -- which would add up to a very different submission, so at this time in my opinion it's not passing the ICLR acceptance threshold.

_______________________________________________________________________________________________________________________________________________________
UPDATE: After reading the rebuttal I think most of my concerns have been addressed and I am updating my score accordingly.

---

> ### Author Response · Authors · 2020-11-19
> **Thank you for your questions and criticisms**
>
> We thank the reviewer for their time and apologize that the initial submission had incomplete experimental details. We have added all such hyperparameter details (regarding data processing, architecture, baselines, optimization) into the Appendix. In the main paper, we have included an Algorithm 1 and Section 3.1 which discusses the general process of how a Neural Event ODE computes piecewise continuous trajectories. We also included an example of the neuronal dynamics model (one of the discrete control environments) in Figure 5 to give readers a visual illustration. In light of this, we hope the reviewer can reassess our paper.
>
> Regarding the experiments, we simply wanted to show that there are numerous application areas. As proof of concept, we constructed problems where we can systematically compare against neural ODEs without event handling and the REINFORCE gradient estimator for temporal point processes. We specifically chose setups such as switching linear dynamical systems, physics and discrete control because these are active lines of research, where similar problem instances are regularly used for comparison and illustration. Existing works on applying neural ODEs to physics-based settings only discuss spring systems or pendulums, with no contact forces, which are arguably much simpler to model due to smoothness. Our method also has distinct advantages over baselines, such as the ability to model discontinuities which ODE-only models cannot, and the ability to learn *deterministic* discrete-valued control policies. Overall, we agree that our method can be applied to more challenging domains, and we are looking forward to pursuing them.
>
>
> Below, we answer all questions asked by the reviewer:
>
> a) Experimental details for baseline models have now been added to the Appendix.
>
> b) Yes, the dynamics can become more complex as training progresses, though this is a property of the ODE component and tangential to our contribution. Regarding the learning of event functions, we empirically found that the number of events tends to have larger oscillations early during training, and stabilize relatively later in the training, though often at a much smaller value than the largest observed during training.
>
> c) Yes, quite interestingly, the specific neural ODE baseline in the physics experiment can recover near-discontinuous changes to the velocity at collision locations, and thus obtain statistically similar performance to our neural event ODE. However, we find that modeling such sudden changes with a smooth transition makes the ODE extremely stiff and require very small step sizes to simulate. We report in the newly updated Table 2 the number of neural network evaluations used by the neural ODE, and it is more than 10x that of the event-based model leading to a substantial increase in runtime. We would imagine the results would be similar with more bouncing balls, at the cost of increasing model capacity and training time due to having more collisions. While the Neural Event ODE must spend more time due to more events, the Neural ODE will, if it can, have to model each of those collisions with stiff dynamics. We also note that the neural event function could likely benefit from more domain knowledge, such as using distances as features, but we stuck to simple MLPs to showcase the generality of our approach.
>
> d) Experimental details are now added to the Appendix, and we have added an extra figure for visualizing a discrete-valued control in a continuous-time environment.
>
> e) Since (continuous) normalizing flows rely on the drift function being the same for all samples (in order to have a normalized probability density), it does not straightforwardly benefit from having sample (i.e. state)-dependent events. We did not go down this route further at this moment in time, though we agree it could be interesting to think about.
>
> f) The colors visualize what our network would predict to be the next switch state *if an event were to occur at that location*. As such, it doesn’t need to learn the actual event boundaries (those are handled by the event function) but simply output the correct switch state at the event boundaries. (We’ll think about how to make this figure more readily understandable.) This is also a key aspect of being able to easily model discontinuities and approximate discrete switch states; if we were to simply let the switch state change continuously in time, it would degenerate to an ODE and would require the switch variable to change near-instantaneously at the boundaries to be accurate.

---

> > ### Comment · AnonReviewer2 · 2020-11-21
> > **Rebuttal**
> >
> > I thank the authors for addressing most of my concerns by adding the details about architectures, algorithm and extra examples. The updated Table 2 now shows more clearly the advantages of the method proposed over the standard Neural ODE -- which adds value to the set of experimental results.
> >
> > I think the manuscript improved a lot after the rebuttal, and I am updating my score accordingly. Although I would like to point out that as the paper was submitted, I found it to be quite incomplete as pointed out in the original review.  That's mainly why I leave it as "Marginally above acceptance threshold" rather than Accept, and will leave it to the ACs to decide how much paper editing after the submission is acceptable.

---

### Official Review · AnonReviewer3 · 2020-10-29
**A nice extension to Neural ODEs**

**Rating:** 7
**Confidence:** 3

**Review:**

This work provides an extension to the neural ODEs framework to include discrete changes (i.e. switching) in continuous-time dynamics. The authors provide a few examples of such systems (bouncing balls, collisions of particles, discrete control systems) and derive formally the gradients with respect to the unknown switching time (which is a solution to the so-called event function), where a discontinuity (the switch) happens. The authors implement their method in the torchdiffeq library of Chen et al. 2018, and provide an extensive experimental evaluation in this manuscript.

The paper is well written, with very few typos and amazing attention to detail (nice colors, thought-through figures, great content organization). Also, the math is well-explained. Overall, a very good read and a nice contribution. I recommend acceptance.

Just a few points/questions for the authors:
- Q1: How does your method scale as the dimension increases? I am thinking about for instance the experiment of section 4.1: if the dynamics is higher-dimensional (with still with 3 discontinuities in the vector field), is the method still able to infer the correct switching times?
- Q2: Still about Figure 2.. how is it possible in panel (d) that the inferred discontinuity in the flow is not aligned with the discontinuity in the field?
- Q3: I am a bit confused about how you infer successive switching times t*1, t*2 etc.. do you have to specify the number of switching times at the beginning of the inference procedure? Do you take gradients with respect to all the switching times together? Could you please add an explanation in the rebuttal and in your revised version of the paper?
- Clarifications: I would add a few lines explaining what the “adjoint state” is before formula 5.
- typos: Dots at the end of formulas are sometimes missing. End of page 4, “SLDS dynamics..”

---

> ### Author Response · Authors · 2020-11-19
> **Thank you for your questions**
>
> We thank the reviewer for their questions. We found these questions very valuable in updating the paper with better and simpler explanations. To aid our explanations, we made use of the extra 9th page to include an Algorithm 1 and Section 3.1 which describes the general process of how a Neural Event ODE computes piecewise continuous trajectories. An visual example of what one of the discrete control environments looks like is also included (Figure 5).
>
> Q1: While the method can be applied as is to any number of dimensions, the gradients with respect to the event functions depend on the events actually triggering. If the drift function never reaches an event boundary (where the event function == 0), then we get no benefit at all from using event functions. We refer to this behavior as runaway event boundaries in the limitations section.However, this is mainly a problem of the drift function not exploring the space sufficiently enough, not a problem of the usage of event functions. To alleviate this, we’re looking into novel parameterizations that couple the drift and event functions, or modeling events within a latent space.
>
> Q2: To clarify (and which now has been explained more clearly in the updated paper), the Neural Event ODE consists of (i) a drift function that is a weighted combination of linear ODEs, weighted by w, (ii) an event function, (iii) an instantaneous modification to the switch state w.
>
> The third component is only invoked at event locations, and modifies w instantaneously. Furthermore, the value of w does not change between events.
>
> The colors visualize what our network would predict to be the next switch state *if an event were to occur at that location*. As such, it doesn’t need to learn the actual event boundaries (those are handled by the event function) but simply output the correct switch state at the event boundaries. This is why the colors don’t match the boundaries in the current manuscript. (We’ll think about how to make this figure more readily accessible.) This is also a key aspect of being able to easily model discontinuities and approximate discrete switch states; if we were to simply let the switch state change continuously in time, it would degenerate to a simple ODE and would require the switch variable to change near-instantaneously at the boundaries to be accurate. This generally would lead to very stiff dynamics, which we discuss in the updated Table 2.
>
>
> Q3: The events of a neural event ODE occur sequentially. Once an event occurs, the state is modified, and then the dynamics resumes, until the next event or we reach the end of the interval (see the new Algorithm 1). In this formulation, we don’t need to provide supervision to the model regarding how many events there are and can train it fully unsupervised in this regard. Our loss only depends on the piecewise continuous trajectory and the gradient is automatically backpropagated to the event times when these states are modified and to the event function parameters. We apologize for the previous lack of explanations around this, and will make the paper more accessible.

---

> > ### Comment · AnonReviewer3 · 2020-11-24
> > **Thanks!**
> >
> > Thank you for your reply. I think the paper is a very valuable contribution and the additional comments make it more clear and accessible.

---

### Official Review · AnonReviewer1 · 2020-10-29

**Rating:** 7
**Confidence:** 4

**Review:**

This paper presents Neural Event ODEs, a method to extend Neural ODEs for modeling discontinuous dynamics in a continuous-time system.  Neural Event ODEs allow to learn termination criteria dependent on the system's state while being fully differentiable. Experiments on time series and temporal point processes validate the benefits of Neural Event ODEs on discountinuous dynamics.

The paper is well written and relatively easy to follow.
The benefits that Neural Event ODEs provide for modeling discontinuous dynamics already becomes apparent from the formulation of its ODE solver. The simplicity of the approach is another advantage, and I can see many possible applications/use cases that can benefit from such an ODE solver.

Nevertheless, there were a few questions and remarks I had when reading the paper:

- In experiment 1, the setup of the Neural ODE is not clear. In particular:
 - How many switch variables have you defined? In case of 3, have you tested what happens if you specify more (e.g. 4-6) to see how sensitive the model is to this parameter?
 - In the result paragraph, a classifier is mentioned that probably should refer to a classifier over $w$. Is this classifier trained by having a weighted average in $f$, i.e. $\frac{d z(t)}{dt}=\sum_{w=1}^{M} p(w)\left[A^{(w)}z+b^{(w)}\right]$, or how is the exact setup?

- Table 2 evaluates the bouncing ball collision experiment on the mean squared error of the predicted trajectories. Based on those scores, the Neural Event ODE performs slightly worse than Neural ODE which is not clearly discussed in the section. Is the conclusion of Neural Event ODE being able to generalize better in this experiment based on qualitative results? If so, wouldn't have an adversarial metric reflected this result better? Besides the region close to the ground, it is hard to tell which model \textit{generalizes} better.

- In section 6, the issue of minibatching is mentioned. How does this effect the results of Neural Event ODEs shown in section 4 and 5.1? Have those models been trained with batch size 1 in contrast to the other methods, or using methods like gradient accumulation of multiple single-batches (hence only time of training increased)?

- Most experimental setups in the paper clearly require the modeling of discountinuous dynamics. Have you tested Neural Event ODEs on tasks where the system to model does not have obvious discontinuities, for instance flow-based generative modeling? Would you see any potential advantages of your method there?

- Experimental details such as the concrete parameterization of $f$ and hyperparameter values is missing for reproducibility of the results. No supplementary materials have been submitted in which this information could have been outlined.

Overall, I think that the ideas presented in the paper are a valuable contribution to the research community and therefore, I would recommend the paper for acceptance, especially if the points mentioned above can be clarified.

Additional comments: Page 4, third line from the bottom has a double punctuation.

---

> ### Author Response · Authors · 2020-11-19
> **Thank you for these questions**
>
> We thank the reviewer for their time and apologize that the initial submission did not contain the Appendix. The updated version has all such hyperparameter details (regarding data processing, architecture, optimization) in the Appendix. We have also updated parts of the main paper to take into account these suggestions and questions, and made use of the allotted 9th page to add extra explanations and visualizations.
>
> Below, we answer all questions asked by the reviewer:
>
> We specified 3 switch variables. We did not perform systematic tests with more switch states but did anecdotally observe that the model can land in bad local minima in such scenarios. Note that since we relax the switch variable in our parameterization, we actually model an infinite set of linear dynamics. A hard switch like the parameterization we used for our discrete control applications might be more preferable as the number of switch states increase, but we did not experiment with this at the time.
>
> The word “classifier” has been removed from the text. We simply meant that at event locations, we instantaneously set $w$ to be the output of a neural network with softmax outputs, so that $w$ sums to one. This is the instantaneous update function in our new Algorithm 1. To clarify, there is no stochasticity involved (unlike previous works on SLDS), and this instantaneous change is part of the neural event ODE and trained end-to-end from just the MSE loss.
>
> Yes, quite interestingly, the specific neural ODE baseline in the physics experiment can recover near-discontinuous changes to the velocity at collision locations, and thus obtain statistically similar performance to our neural event ODE. However, we find that modeling such sudden changes with a smooth transition makes the ODE extremely stiff and require very small step sizes to simulate. We report in the updated Table 2 the number of neural network evaluations used by the neural ODE baseline, and it is more than 10x that of the event-based model leading to a substantial increase in runtime. We also note that the neural event function could likely benefit from more domain knowledge, such as using distances as features, but we stuck to simple MLPs to showcase the generality of our approach.
>
> To clarify, we do handle multiple event functions (by aggregating them to form one event function). However, the lack of parallelism cancels out some of the benefits of batching: since minibatching roughly linearly increases the number of events, our ODE solver (because it has no notion that some events are independent) restarts the local extrapolation for the entire minibatch even when only one event function from the minibatch is triggered. This is mainly a software issue, albeit with no straightforward solution, and we did not find that the extra gradient signal from batching helped train any faster in terms of wallclock time. We plan on tackling this more as we scale up to bigger applications where batching is more effective.
>
> Since (continuous) normalizing flows rely on the drift function being the same for all samples (in order to have a normalized probability density), it does not straightforwardly benefit from having sample-dependent events. We did not explore this further at this moment in time, though we agree it would be interesting to think of novel applications where events may be beneficial. We did not test neural event ODEs on tasks where event functions would not be beneficial, though in such cases we imagine the event functions may trigger, leading to extra processing overhead, while the instantaneous updates that introduce discontinuities are effectively reduced to zero.

---

> > ### Comment · AnonReviewer1 · 2020-11-19
> > **Thank you for the response**
> >
> > Thank you for clarifying all the questions above. It would be indeed interesting to see the approach being scaled up and applied to more complex domains. But given the scope and other contributions of the paper, it is reasonable to leave this for future work.

---

### Decision · Program_Chairs · 2021-01-07
**Final Decision**

**Decision:**

Accept (Poster)

**Comment:**

This paper presents an extension of the neural ODE approach to include discrete changes in the continuous-time dynamics. All reviewers agree the contribution made by this paper is worth publishing. Most of the reviewers' concerns have been answered in the rebuttal and I therefore recommend accepting this paper.